# A Prospective Social Life Cycle Assessment (sLCA) of Electricity Generation from Municipal Solid Waste in Nigeria

Oluwaseun Nubi *, Stephen Morse and Richard J. Murphy

Centre for Environment and Sustainability, University of Surrey, Guildford GU2 7XH, UK;
s.morse@surrey.ac.uk (S.M.); rj.murphy@surrey.ac.uk (R.J.M.)
* Correspondence: o.nubi@surrey.ac.uk

**Abstract:** This research assesses the social impacts that could arise from the potential waste-to-energy (WtE) generation of electricity from municipal solid waste (MSW) in the cities of Lagos and Abuja in Nigeria. Social life cycle assessment (sLCA) was the main analytical approach used coupled with a participatory approach to identify relevant social issues to serve as the potential sLCA impact 'subcategories'. Focus group research in both cities led to the identification of 11 social issues that were transformed into social impact subcategories with appropriate indicators for the sLCA. These were populated with data from a questionnaire-based survey with approximately 140 stakeholders. The results indicated that the impact subcategories "*Improved Electricity Supply*" and "*Income*" were ranked respectively as having the most and the least significant social impacts associated with the potential adoption of WtE in these two cities in Nigeria. Overall, the research showed that the expected social impact was higher for WtE electricity generation in Lagos than in Abuja. This difference may be related to the higher population and greater amounts of waste in Lagos and its position as a hub for many of the country's commercial and industrial activities which have long been affected by inadequate electricity supply. This study also provides an example of the use of participatory processes as an important approach in sLCA for the elucidation of social issues that are directly pertinent to key local perspectives when considering such technology implementations.

**Keywords:** waste-to-energy; social life cycle assessment; participatory approach; indicators

## 1. Introduction

Population growth, changes in consumption patterns, increased urbanization, industrialization, and economic growth are associated with increased rates of municipal solid waste (MSW) generation and the increased demand for energy, particularly electricity, in developing countries such as Nigeria [1]. The effective management of waste and a reliable energy (electricity) supply are vital to the development and living standards of any nation [2]. However, deficiencies in these services have been a long-running and acute problem in some countries [3]. For example, in Nigeria, the population is estimated to be close to 200 million people who produce about 32 million tonnes of MSW annually [4]. This equates to a generation rate of 0.438 kg/person/day but with a collection rate estimated to be only between 20% and 40% [4]. In addition, the average per capita electrical energy consumption in Nigeria is 134 kWh per annum and the power generation installed capacity of the national grid in 2017 was 12 GW [2]. These are well below those of other developing countries, e.g., South Africa (4363 kWh, 47 GW) and Indonesia (609 kWh, 53 GW) for the same year and suggests a substantial power deficit in Nigeria [2]. One option to help address the twin concerns of weak management of MSW and an inadequate electricity generation capacity is the use of MSW for power (and heat) generation, commonly referred to as waste-to-energy (WtE). WtE involves the use of thermal and biological processes to extract energy stored in several components of MSW and convert it to heat or electricity or in the case of combined heat and power (CHP) into both [5]. Incineration, gasification, and

pyrolysis processes represent the 'thermal' route to electricity generation while processes such as anaerobic digestion and landfill gas recovery are commonly termed the 'biological' route [6]. The choice as to which technology to use may be influenced by parameters such as the moisture content and calorific value of the waste as well as the cost of installation. For instance, incineration is one of the oldest methods in WtE management.

This process has the capability to decrease the volume of MSW by as much as 80–90% and is most suitable for high calorific value waste but is not well suited for waste with a high moisture content, low calorific value, and chlorinated waste [7]. In addition, the capital and compliance costs are medium to high because of the need for heavy equipment (e.g., furnace) and skilled staff [8].

There is extensive literature on the environmental impact of many WtE technologies in both developed and developing countries. For example, Ayodele et al. [9] evaluated the environmental sustainability (via the impacts on global warming potential, acidification potential, carcinogenic reduction potential, etc.) for WtE technologies for 12 cities in Nigeria to determine the option best suited to the locations. They found that a hybrid of incineration/anaerobic digestion was the best option when compared with the other methods in terms of global warming potential and acidification potential while landfill gas to energy technology was the best in terms of carcinogenic reduction potential. Gunamantha and Sarto [10] carried out an environmental analysis to compare various energetic valorization options in Indonesia and concluded that direct gasification had the best environmental profile for most of the impact categories assessed (except acidification). In both studies, the common analytical tool used to assess the environmental impacts was life cycle assessment (LCA). However, there have been relatively few published analyses of the social impacts of WtE [11,12]. While communities near waste sites can be affected by WtE either positively or negatively, others may make a living from MSW, either via direct employment or as entrepreneur 'pickers' (scavengers) who select waste that they can sell [13]. Indeed, there can be a significant overlap in the materials collected by 'pickers' and those seeking WtE initiatives [13]. Thus, more work is required on the social impacts of WtE, especially within the developing world context [11].

The research reported here aimed to explore the potential social impacts of WtE using MSW in two urban areas (Lagos and Abuja) of Nigeria. The approach taken was that of social life cycle assessment (sLCA) with inputs from those directly engaged in the MSW management and energy sectors as well as members of the general public. sLCA, as a tool used to assess the social aspect of sustainability, differs from approaches that have been used previously to assess the social impacts of WtE [14]. The rationale for adopting sLCA in the present research is given later in the paper (see Section 2.2), but it forms part of a growing interest in the application of sLCA in various contexts [15–17]. In addition, it has a distinct advantage in that it forms a core component of the integrated sustainability assessment framework of the life cycle sustainability assessment (LCSA). LCSA is attracting much attention as it seeks to combine environment LCA (eLCA), life cycle costing (LCC), and sLCA within a unified assessment [18], via guidelines such as those of [19–22]. sLCA typically involves the use of a suite of indicators to assess social impacts that are then classified into various 'impact categories' [21,22]. The selection of appropriate indicators for social impact is a key step in sLCA and there have been calls for the adoption of participatory approaches to this indicator selection [23]. However, while these approaches have advantages as they encourage expression, communication, and build agreement, much can depend on who is involved in the process [23]. In the present study of Lagos and Abuja in Nigeria (two cities with rather different characteristics, see below), we considered that the participatory process would be valuable in revealing potentially differing perceptions of the social impacts of WtE in their respective local contexts.

Nigeria has yet to adopt WtE, but there are four technologies that are well-established globally (incineration, anaerobic digestion, gasification, and landfill gas to energy technology) and, if Nigeria were to go down the WtE route, then one or more of these is likely to be used. It should be noted that this research presents a prospective, future-looking sLCA on WtE in general, rather than seeking to make direct comparisons between differing specific WtE technologies. This is because (i) such comparisons would be limited by the very sparse knowledge and experience of different WtE technologies in Nigeria at present and, (ii) it has been noted in sLCA practice that it is often challenging to attribute social impacts directly to specific processes or products [24].

Furthermore, there also remains a question as to how best to determine specific process/product attribution in a prospective " ... *what if WtE were to be implemented* ... " study such as this. Thus, making a valid, direct comparison between the social impacts of these technologies was considered premature.

Our overall aim in this research was to determine the extent to which local stakeholders might hold common or differing views on the social impacts of implementing WtE in these two major cities in Nigeria. It was conducted to reveal anticipated social benefits or otherwise that may attend efforts to seek general WtE solutions to alleviate the challenges that apply to the electricity generation and the organized waste management systems of Nigeria. This sLCA study forms part of an ongoing wider, integrated assessment research study of the potential for WtE in Nigeria.

## 2. Materials and Methods

### 2.1. Lagos and Abuja

Lagos is located on the southwestern coast of Nigeria, with a land area of 3577 km$^2$ and a population of approximately 21 million people. The city serves as the primary commercial center of Nigeria [25]. The city generates waste at 0.72 kg/person/day which is approximately 15,000 tonnes of waste daily [26]. With a population growth rate of 3.26%, the quantity of waste is expected to keep increasing [27]. Lagos also has many production and service industries in the formal and informal sectors that account for more than 70% of the city's urban economy [28], which in turn generate a significant amount of waste. Hence, there is a need for efficient and cost-effective management of MSW from both domestic and industrial sources [29], and the authorities have taken various steps to address waste management issues, primarily via the Lagos State Waste Management Authority (LAWMA) [30].

LAWMA is responsible for collecting waste from public areas, while private companies, under a PSP (Private Sector Participation) programme, collect waste from residential and commercial areas [26]. LAWMA collects solid waste from public areas daily while PSP companies collect waste once or twice weekly and adopt a door-to-door collection method [31]. However, the coverage of waste collection is about 63% of the city's area while the remaining 37% of the city's area (mostly the outskirts) is sparingly, if ever, serviced [26]. LAWMA operates four major landfills in the city (Figure 1) and these include the Abule-Egba landfill site, the Olushosun landfill site, and the Solous I and II landfill sites [32]. It is these landfill sites that receive most of the waste generated in Lagos and almost all of them are now nearing their planned maximum capacity.

As with much of Nigeria, Lagos is also confronted with an inadequate supply of electricity. An energy demand analysis carried out by the Lagos State Government showed that the total demand for electricity in Lagos was estimated to be 10 GW in 2011 and 11 GW in 2015, with 70% of this needed for residential use [34]. However, a severe shortfall in energy supply to the city leaves its residents with just 1 to 5 h of electricity daily [35], and industries based in Lagos have little choice but to purchase standby generators [34].

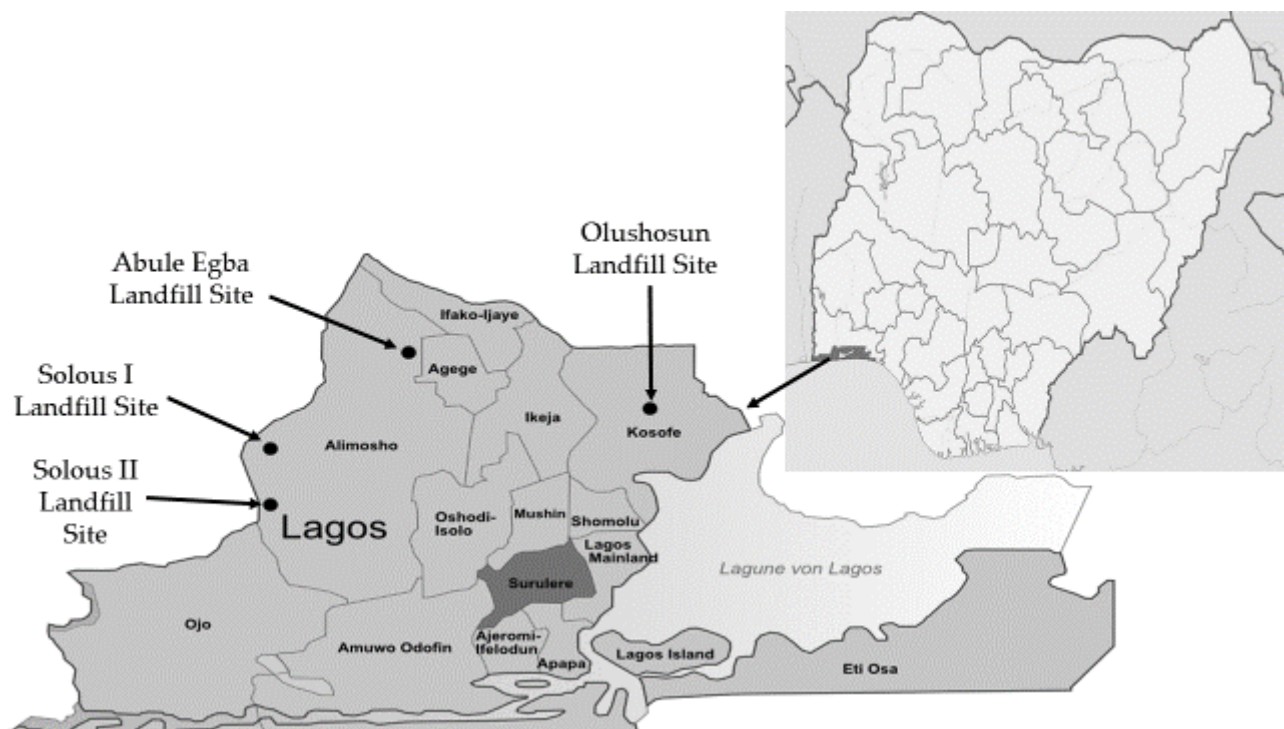

**Figure 1.** Map of the Lagos metropolitan area indicating the major landfill sites [33].

Abuja, the administrative and political capital of Nigeria, is located in the geographical center of the country, within the Federal Capital Territory (FCT). The city has a population of approximately 6 million people and a land area of 1769 km² [36]. According to a 2014 report from the Federal Ministry of Environment, the city generates about 3390 tonnes of MSW daily with an average waste generation rate of about 0.55–0.58 kg/person/day [37]. As with Lagos, it is expected that the amount of waste generated by the city will increase significantly [38]. The Abuja Environmental Protection Board (AEPB) is charged with the responsibility of managing MSW in the city and has attempted to achieve this by setting up a PSP program as well, with many individual companies which were allotted districts for collecting and transporting waste to various disposal sites in the city [39].

There are four major waste disposal sites managed by AEPB, located in Mpape, Gousa, Ajata, and Kubwa (Figure 2), and unlike Lagos, these are mostly located well outside the city. However, the Mpape, Ajata, and Kubwa disposal sites were closed in 2005 due to problems with odour, air pollution, and fire outbreaks [36]. Presently, the MSW is transported to a single dumpsite at Gousa, and this site is almost at full capacity [40].

As with Lagos, electricity demand in Abuja far outweighs the current supply [41], and as a result, there has been a huge shortage in the city's power supply which has been attributed to load shedding from the national grid. Abuja is reported to require about 400 to 500 MW per day but is supplied with between 200 and 300 MW of electricity per day [41]. Like Lagos, households and companies attempt to bridge this gap in supply by using alternative sources such as diesel generators [42].

Both cities share similar problems associated with rapid population and economic expansion, especially in terms of waste management and electricity supply. However, the cities also differ in some regards. Lagos is much older than Abuja, has a higher population size and density, and spans a larger land area. Lagos is hemmed in by the ocean at one side and its development has largely been unplanned, with dwellings packed densely together and streets typically being narrow and choked with traffic. Landfill sites in Lagos are often surrounded by dense urban development. Abuja, by contrast, has the advantage of being a planned city with wide streets laid out in a grid pattern which facilitates transport. Being at the geographical center of the country also provides space for the development of the city

and its landfill sites are outside the city. In terms of waste, the MSW composition in both cities has a high organic content probably due to a preponderance of food waste [43]. This could aid the adoption of WtE, but challenges such as inadequate funding for such projects and lack of political will from the government have not encouraged its development [44]. Nonetheless, Emmanuel et al. [45] noted that WtE has the capacity to generate sufficient electricity to power over 11,000 and 94,000 homes in Abuja and Lagos, respectively, as well as facilitating improved waste management. Thus, while both cities share similar issues of waste and energy, the differences between them may also result in differing social impacts arising from WtE.

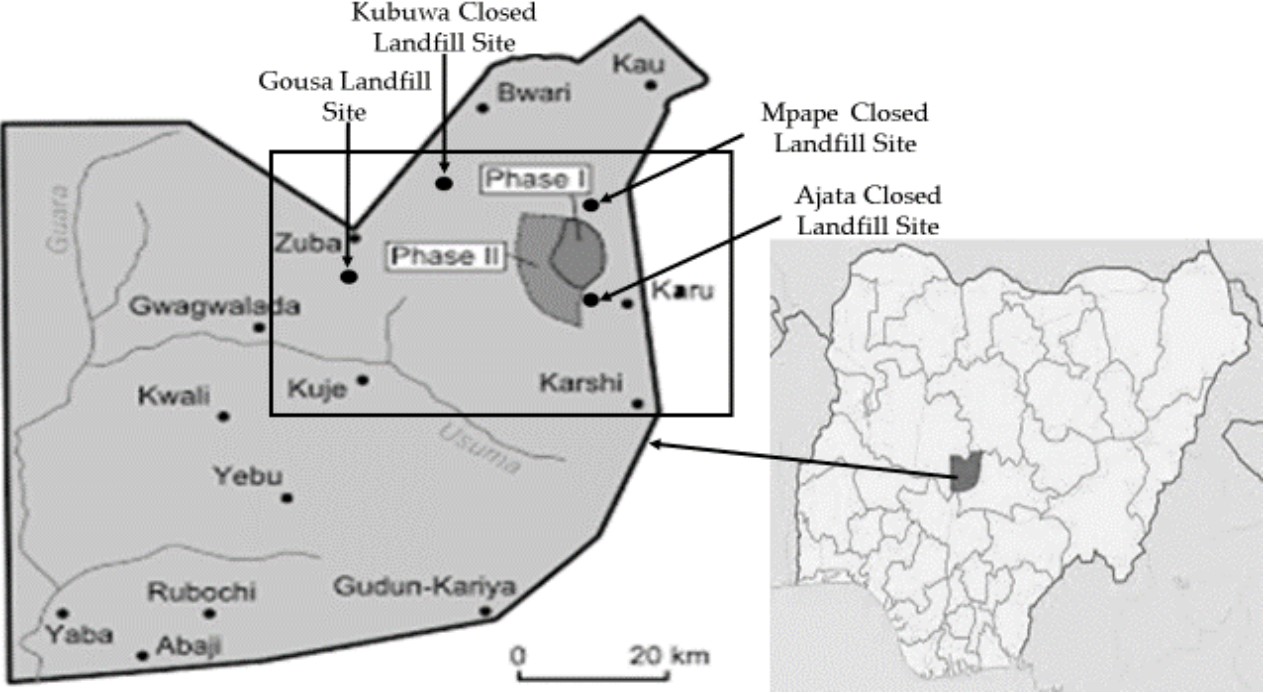

**Figure 2.** Map of the Federal Capital Territory showing the approximate location of the Abuja metropolitan area (east of Gwagwalada) with the major landfill sites [33].

### 2.2. sLCA—Goal and Scope Definition and Functional Unit

The guidelines for sLCA from UNEP (and previously from UNEP-SETAC) follow largely a 'classical' LCA structure with the definition of goal and scope, functional unit declaration, life cycle inventory, and life cycle impact assessment and interpretation [21,22]. In the research reported here, our goal was to carry out a prospective sLCA of WtE in Lagos and Abuja and the scope was based on the UNEP 'Social Impact Subcategories'.

The functional unit for the study was to assess the social impact arising from a potential shift in the management of MSW in future years and over an indefinite time frame and can be defined as:

> "the prospective management of MSW for WtE electrical power generation in Lagos and Abuja, Nigeria"

Social Impact categories refer to logical groupings of impacts in relation to social issues of interest to policymakers and managers [21,22]. The subcategories refer to socially relevant characteristics or attributes that can be assessed (via indicators), of which several may be used to cover an impact category [21,22]. The indicators provide the most direct evidence of the condition or the result of the impact categories being measured [21,22] and have characteristics such as type (qualitative, quantitative, or semi-quantitative) and unit of measurement. It is possible to have weighting/aggregation steps, which allow the passing from inventory indicator results to a subcategory result and, in turn, from subcategories to

an impact category result. The framework of indicators, subcategories, and categories can be set by an external body, but there is a growing body of research [46], which suggests that they can also be defined via a participatory process which was the approach taken in this study. Hence, in this study, a series of focus groups in Lagos and Abuja were used to select the 'Social Impact subcategories' and provide insights regarding the choice of indicators.

### 2.3. Data Collection

The first stage of data collection took the form of a series of focus groups designed to explore various stakeholders' views on the main social issues associated with the adoption of WtE (Table 1). Four groups were held in Lagos, two in Abuja, with the difference in numbers mainly due to logistical and attendee availability constraints. Participants in the focus groups comprised members of the public, staff of the government waste management authorities, members of the private waste management sector, academics/researchers, and members of the electricity sector. In the UNEP guidelines, the use of six stakeholder groups (workers, local community, society, consumers, value chain actors, and children) is suggested, although it is noted that the list is not comprehensive and requires further development [22,47]. In this study, the participants came under the following UNEP categories:

- Workers: people working in the formal and informal sectors of waste management services,
- Local Community: individuals living near the dumpsite and possibly the WtE plant.
- Society: those who are indirectly affected by the quality of the implementation of WtE technology,
- Consumers: people receiving waste management services and electricity supply.

**Table 1.** Participant composition of focus groups (stakeholder categories based on [21,22]).

| City | Focus Group Number | Number of Participants | Types of Participant | UNEP/SETAC Category |
|---|---|---|---|---|
| Lagos | 1 | 10 | Waste Pickers | Workers/Local Community |
| | | | General public | Consumers |
| | 2 | 7 | Academics/Researchers | Society |
| | 3 | 7 | Lagos State Waste Management Authority | Workers |
| | 4 | 8 | Private Waste Collection Sector | Workers |
| Abuja | 1 | 8 | Abuja Environmental Protection Board | Workers |
| | | | Private Waste Collection Sector | Workers |
| | | | Nigeria Electricity Regulatory Commission | Society |
| | | | General Public | Consumers |
| | 2 | 7 | Abuja Environmental Protection Board | Workers |
| | | | Private Waste Collection Sector | Workers |
| | | | Abuja Electricity Distribution Company | Society |
| | | | General Public | Consumers |

The selection of these relevant stakeholder groups for this study was based on a literature review.

Each focus group began with an 'ice-breaker' exercise and a brief technical introduction to the four WtE technologies that would, if the country opted to do so, be those most likely to be adopted. Participants were then divided into groups by the researcher (lead author) and tasked with identifying what they considered to be key social issues that might be associated with the adoption of WtE in their cities and would provide relevant local perspectives on which sLCA social impact subcategories could be based for use in this study. The combined lists of social issues emerging from the focus groups were then reviewed in relation to the 'standard' subcategories recommended in the UNEP guidelines [21,22], and indicators to calibrate these and additional impact subcategories, the focus groups identified in this study were assigned by the researchers (see Table 2).

The indicators for subcategories such as '*Improved Sanitation*' and '*Improved Electricity Supply*' that were identified from the participatory research with a focus group (and not found in the UNEP guidelines) were based on a literature review relating to social issues (e.g., benefits, challenges, problems) concerning the implementation of WtE.

For the subcategory '*Income*', the indicators used were based on the estimation of how the introduction of WtE could impact disposable income (the difference between the income earned and expenditure incurred).

### 2.4. Inventory Analysis

The second stage of the sLCA involved an inventory analysis whereby indicators are 'populated' with appropriate data (i.e., given values). In this study, the process was achieved via a combination of face-to-face interviews and an online questionnaire-based survey with 70 key informants in Lagos and 65 in Abuja (Table 3). The selection of the informants was carried out by convenience sampling. Table 3 shows that the majority of the respondents in both cities have bachelor's degrees. The largest proportion of respondents were within the age ranges of 41–50 followed by those in the 20–30 and 31–40 age brackets. The monthly earnings of the largest group of respondents in Lagos were USD 300 and above, while most in Abuja earn between US$ 200 and US$ 300 per month. In Lagos, a higher percentage of respondents worked in the private sector, the reverse being the case in Abuja. Both cities, however, showed that the number of male respondents was higher than females.

**Table 2.** Social impact subcategories and indicators used, with the relevant stakeholders identified from the focus groups and literature (W—Workers, C—Consumers, LC—Community and S—Society). (*) is used to denote those Social Impact Subcategories in the study that correspond to those in the UNEP Guidelines.

| Social Impact Subcategory | Indicator |
| --- | --- |
| 1. Employment * | 1.1 Number of Jobs created [W] |
| | 1.2 Level of Job Creation Opportunities by WtE [W] |
| 2. Public Awareness | 2.1 Level of Awareness of the Electricity Generation from MSW [C], [W], [S] and [LC] |
| 3. Health and Safety * | 3.1 Level of expected Accidents/injuries/fatalities [W] |
| | 3.2 Occupational health risk perception [W] |
| | 3.3 Health and Safety awareness [W] |
| | 3.4 Safety Risk perception of the WtE system [W] |
| | 3.5 Protective Equipment availability [W] |
| | 3.6 Effect of WtE service on local community's health and safe living condition [LC] |
| | 3.7 Endangerment of WtE service for the local community's secure living condition [LC] |
| | 3.8 Effect of air pollution of a WtE plant on Local Community [LC] |
| | 3.9 Effect of water pollution of a WtE plant on Local Community [LC] |
| | 3.10 Effect of land pollution of a WtE plant on Local Community [LC] |
| | 3.11 Effect of noise pollution of a WtE plant on Local Community [LC] |



**Table 2.** *Cont.*

| Social Impact Subcategory | Indicator |
|---|---|
| 4. Location | 4.1 Effect of proximity of a WtE plant on local residences in terms of public health concerns [LC] |
| | 4.2 Effect of proximity of a WtE plant on local residences in terms of significant environmental problems [LC] |
| | 4.3 Effect of the proximity of a WtE plant on local residences in terms of sale or rent of properties [LC] |
| | 4.4 Effect of the proximity of a WtE plant on local residences in terms of promoting economic/commercial activities [LC] |
| | 4.5 Effect of the proximity of a WtE plant on local residence in terms of encouraging job creation [LC] |
| | 4.6 Level of comfort of having a WtE plant within local vicinity [LC] |
| | 4.7 Waste Management as the most critical factor in the implementation of a WtE plant [LC] |
| | 4.8 Electricity Production as the most critical factor in the implementation of WtE plant [LC] |
| | 4.9 Environmental Pollution as the most critical factor in the implementation of a WtE plant [LC] |
| | 4.10 Aesthetics as the most critical factor in the implementation of a WtE plant [LC] |
| | 4.11 Local Traffic burden as the most critical factor in the implementation of a Waste to Energy plant [LC] |
| | 4.12 Job creation as the most critical factor in the implementation of WtE plant [LC] |
| | 4.13 Safety of WtE plant on Public Health [LC] |
| | 4.14 Opposition to the Construction of WtE plant in town/region [LC] |
| 5. Contribution of Waste Management to Economic Development * | 5.1 Percentage Contribution of Waste Management to the GDP of Nigeria [S] |
| | 5.2 Contribution Level of WtE to Economic Development [S] |
| | 5.3 Amount of revenue generated annually generated from waste management (US$) [S] |
| 6. Public Acceptance | 6.1 Extent of public acceptance for WtE technology [C], [W], [LC] and [S] |
| 7. Government Policy | 7.1 Extent of Strategic Action on policies on waste management/WtE [W] |
| | 7.2 Adequacy of laws regulating waste management [W] |
| | 7.3 Strength of waste management institutions [W] |
| 8. Education and Training * | 8.1 Adequacy of public education on waste management [W] |
| | 8.2 Adequacy of level of training of waste workers [W] |
| 9. Improved Sanitation | 9.1 Implementation and Access to Improved Sanitation due to WtE [C] |
| | 9.2 Encouragement in the participation in waste sorting and other sanitation exercise due to WtE [C] |
| | 9.3 Effect on the payment for sanitation services due to WtE [C] |
| 10. Improved Electricity Supply | 10.1 Amount of Electricity Supplied Annually (kWh) [S] |
| | 10.2 Extent of improvement in electricity supply [S] |
| 11. Income | 11.1 Impact of WtE on Monthly Income of Workers (US$) [W] |
| | 11.2 Impact of WtE on Monthly Income of Consumers (US$) [C] |
| | 11.3 Impact of WtE on Paid Living Wage [W] |
| | 11.4 Impact of WtE on Regular Payment [W] |

* indicates a subcategory in the UNEP Guidelines—all other subcategories were developed for this study based on the issues identified by the local Focus groups.

**Table 3.** Breakdown of respondents in the questionnaire-based survey for Lagos and Abuja.

| | Number of Respondents By Category | |
| --- | --- | --- |
| | **Lagos** | **Abuja** |
| Stakeholder Groups | | |
| Workers | 20 (29%) | 20 (31%) |
| Local Community | 10 (14%) | 10 (15%) |
| Society | 10 (14%) | 10 (15%) |
| Consumers | 30 (43%) | 25 (39%) |
| Gender | | |
| Male | 47 (67%) | 43 (66%) |
| Female | 23 (33%) | 22 (34%) |
| Age | | |
| 20–30 | 20 (29%) | 19 (29%) |
| 31–40 | 16 (23%) | 14 (22%) |
| 41–50 | 32 (45%) | 30 (46%) |
| 50 and Above | 2 (3%) | 2 (3%) |
| Educational Qualification | | |
| Primary School Certificate | 5 (7%) | 7 (11%) |
| Secondary School Certificate | 10 (14%) | 21 (32%) |
| Bachelor's Degree | 35 (50%) | 23 (35%) |
| Master's Degree | 16 (23%) | 13 (20%) |
| Doctorate Degree | 4 (6%) | 1 (2%) |
| Monthly Income Level (US$) | | |
| Less than 200 | 6 (9%) | 15 (23%) |
| 200–300 | 31 (44%) | 26 (40%) |
| 300 and Above | 33 (47%) | 24 (37%) |
| Occupational Sector | | |
| Public | 30 (43%) | 45 (62%) |
| Private | 40 (57%) | 25 (38%) |
| Total (all respondents by city) | 70 (100%) | 65(100%) |

Each interview was conducted in English and was based on a questionnaire comprising about 40 standardized questions linked to the indicators associated with the issues that emerged from the participatory focus group process (Table 2). It can be challenging to obtain values for the indicators within such 'what if' contexts and previous studies have often relied on using scores for assessing social impact. For instance, Spillemaeckers et al. [48] applied a binary scale (0 or 1) to assess the fulfillment of social indicators, whereas Umair et al. [49] characterized data as either negative (−) or positive (+). Blom and Solmar [50], on the other hand, used a scale of 1, 0, and −1 where the figures represent a negative, the absence of any issues (or data), and positive social impact, respectively. A similar scoring system was used by Wan [51], but with the scale reversed such that +1 and −1 referred to positive and negative social effects, respectively. Foolmaun and Ramjeeawon [52] suggested the use of a logical scoring system where indicator results are first converted to percentages and then placed into five score categories: 0–20% (1), 20–40% (2), 40–60% (3), 60–80% (4), and 80–100% (5). Ciroth and Franze [53] proposed a six-scale scoring system in which social performance was classified from poor to excellent.

In this study, the respondents were asked to provide a value for the indicators using a five-point Likert-type scale. The indicators were expressed in the form of questions with respondents being able to answer 1 = Very Low, 2 = Low, 3 = Medium, 4 = High, and 5 = Very High. For example, under the 'Employment' sub-category the respondents were asked to "Rate the level WtE will influence job opportunities". The responses were converted into sLCA inventory data through a characterization method whereby scores from the Likert scale were converted into impact scores [16]. For categories within a single

indicator, such as "*Public Awareness*" and "*Public Acceptance*", each indicator Likert score was multiplied by the respective number of respondents and the values aggregated. The total was then divided by the total number of respondents. This produced the impact score for the social impact subcategory. In cases where a social impact subcategory had more than one indicator, the scores obtained for each indicator (derived as described above for the single indicator cases), were summed and divided by the number of indicators to obtain an average overall 'Social Impact' score for the subcategory. It should be noted that all indicators within each subcategory were given equal weight. For the numerical indicators used for the Social Impact subcategories, such as "*Employment*", "*Contribution of Waste Management to Economic Development*", and "*Improved Electricity Supply*" the numerical data were classified into one of the five 'bands' indicated in Table 4 to generate scores following the approach of [52]. This conversion allowed the values to be handled in the same way as other indicators in the same social impact subcategories which used Likert-type scales. Statistical analysis (using the Kruskal–Wallis non-parametric test) was performed to test for any significant differences in social impact subcategories between cities.

**Table 4.** Banding used for the indicators in subcategories with numerical data.

| Indicator | Scores Assigned for 'Bands' of Numerical Data | | | | |
| | 1 | 2 | 3 | 4 | 5 |
|---|---|---|---|---|---|
| Number of Jobs Created in Waste Management Sector | 0–10,000 | 10,001–20,000 | 20,001–30,000 | 30,001–40,000 | 40,001–50,000 |
| Contribution of the Waste Management Sector to Economic Development (% of GDP) | 0–1.49% | 1.5–2.49% | 2.5–3.49% | 3.5–4.49% | 4.5–5% |
| Annual Revenue Generated from Waste Management (US$) | 0–10,000,000 | 10,000,001–20,000,000 | 20,000,001–30,000,000 | 30,000,001–40,000,000 | 40,000,001–50,000,000 |

The final step in the process was to use color coding to characterize the 'Social Impact' of each subcategory as Very-Negative [Red] (1–1.5), Negative [Orange] (1.5–2.5), Neutral [Yellow] (2.5–3.5), Positive [Light Green] (3.5–4.5), and Very-Positive [Green] (4.5–5) [16] (Table 5).

**Table 5.** Characterization of impact in the Social Impact subcategories.

| Colour Code for Social Impact Subcategory | Scores |
|---|---|
| Very Negative | 1.0–1.5 |
| Negative | 1.5–2.5 |
| Neutral | 2.5–3.5 |
| Positive | 3.5–4.5 |
| Very Positive | 4.5–5.0 |

An example of data processing is provided below for the "Public Awareness" subcategory in Lagos:

Step 1 Derivation of Overall Score for Indicators: The individual scores given by respondents for the indicator were converted to overall means (with standard deviations) via an averaging process of multiplying the number of respondents choosing a particular score value (between 1 and 5) by that value and then dividing the total by the number of respondents. For example, with responses to the question (i.e., indicator) "*Level of Awareness about the Electricity Generation from MSW*" in Lagos:

$$\text{Total Likert Score} = \sum \text{No. respondents with Likert Scale Scores of 1 to 5} \times \text{Score Values}$$
$$\text{Likert Score 5 (Very High)} = 39 \times 5 = 195$$
$$\text{Likert Score 4 (High)} = 5 \times 4 = 20$$
$$\text{Likert Score 3 (Medium)} = 2 \times 3 = 6$$
$$\text{Likert Score 2 (Low)} = 16 \times 2 = 32$$
$$\text{Likert Score 1 (Very Low)} = 8 \times 1 = 8$$

(1)

Step 2 Summation and Averaging of Scores: All the scores are summed to obtain a total score for the indicator. An average score for the indicator was obtained by dividing the total score by the total number of respondents for each stakeholder.

$$\text{Indicator Score } = \frac{(\text{Total Likert Score})}{\text{Total Number of Respondents}} \text{Indicator Score } = \frac{(195 + 20 + 6 + 32 + 8)}{70} = \frac{261}{70} = 3.73 \quad (2)$$

Step 3 Conversion to Social Impact scores: The individual indicator scores were either used directly when there was only one indicator per subcategory as for "Public Awareness" or, in cases where there were multiple indicators, the individual indicator scores were summed and averaged to obtain a single value for the Social Impact subcategory, thereby allowing for comparison between subcategories.

$$\text{Impact Score for Public Awareness } = \frac{\sum(\text{Total Indicator Score})}{\text{Total Number of Indicators}} = \frac{3.73}{1} = 3.73 \quad (3)$$

This Score indicates a 'Positive' Social Impact (Table 5) for the subcategory "Public Awareness".

### 3. Results

*3.1. Scoring of Social Impact Indicators*

Table 6 presents the percentage of respondents giving Likert-scale scores for each indicator, with an overall average score for each city. Table 7 gives the scores assigned for those indicators populated using numerical raw data converted to the five bands for each city.

Table 7 shows the social impact indicators with their numerical data. For the indicators "*Number of jobs created*" and "*Amount of revenue generated from waste management*", Lagos had substantially higher values than Abuja. The reverse was true for the indicator "*Amount of Electricity Supplied Annually*" which was higher for Abuja than for Lagos (in that year).

*3.2. Overall Social Impact of Social Impact Sub-Categories*

Table 8 shows the combined indicator scores and color codes for each Social Impact subcategory. Statistical analyses (Kruskal–Wallis test) were performed to determine significant differences in the subcategories between the two cities. Significant differences in the subcategories between Lagos and Abuja were found for all social impact subcategories, except for "*Employment*" and "*Location*".

There was also a tendency across the 11 categories for respondents from Lagos to score subcategories more highly than respondents from Abuja and this is borne out in the overall average score for the whole set of subcategories with Lagos at 3.97 (sd 0.34) and Abuja at 3.23 (sd 0.56).

In terms of the overall combined scores for both cities, "*Improved Electricity Supply*" (score 4.26), "*Employment*" (4.10), "*Contribution of Waste Management to Economic Development*" (4.00) and "*Public Acceptance*" (3.92) were identified as having the highest impact scores out of the 11 Social Impact subcategories evaluated. "*Improved Electricity Supply*" received the highest overall combined score and the highest individual score of social impact in Lagos and the second highest individual score for Abuja.

**Table 6.** Scoring of Social Impact Indicators—Likert scale assessment [percentage of respondents assigning Likert scale scores (1–5) per indicator. Average subcategory scores in the right-hand column (upper values = Lagos, lower values = Abuja)].

| Social Impact Subcategory | Question/Indicator | Proportion (%) of Respondent Scores per Indicator for Likert Scale Scores (Highest Values Indicated in Bold) and Associated Band Score for Indicators from Numerical data | | | | | Subcategory Average Score |
|---|---|---|---|---|---|---|---|
| | | 1 | 2 | 3 | 4 | 5 | |
| | | Very Low | Low | Medium | High | Very High | Lagos Upper |
| | | | | | | | Abuja Lower |
| Employment | Rate the level WtE will create more job opportunities | 0 / 5 | 3 / 3 | 23 / 12 | 31 / 43 | 43 / 37 | 4.14 / 4.05 |
| Public Awareness | Rate the level of awareness of electricity generation from MSW | 11 / 12 | 23 / 19 | 3 / 52 | 7 / 17 | 56 / 0 | 3.73 / 2.74 |
| Health and Safety | Rate the expected occurrence of accidents/injuries/fatalities arising from the introduction of WtE | 9 / 11 | 19 / 22 | 6 / 49 | 39 / 12 | 29 / 6 | 3.60 / 2.82 |
| | Rate the existence of occupational health risk associated with WtE | 0 / 6 | 9 / 25 | 41 / 37 | 17 / 28 | 33 / 5 | 3.74 / 3.00 |
| | Rate the level of health and safety awareness | 0 / 11 | 26 / 15 | 31 / 31 | 19 / 28 | 24 / 15 | 3.41 / 3.22 |
| | Rate the existence of safety risk in the system | 0 / 14 | 44 / 22 | 3 / 34 | 26 / 19 | 27 / 12 | 3.36 / 2.94 |
| | Rate the presence of protective equipment with the introduction of WtE | 14 / 6 | 0 / 34 | 41 / 34 | 27 / 12 | 17 / 14 | 3.33 / 2.94 |
| | Rate the effect of the service on the local community's health and safe living conditions. | 17 / 11 | 20 / 31 | 16 / 34 | 21 / 19 | 26 / 6 | 3.19 / 2.78 |
| | Rate the endangerment of the service on the local community's secure living conditions. | 14 / 6 | 26 / 34 | 13 / 46 | 21 / 14 | 26 / 0 | 3.19 / 2.68 |
| | Rate the effect of air pollution when building a WtE Plant | 0 / 14 | 3 / 34 | 34 / 31 | 14 / 19 | 49 / 3 | 4.09 / 2.63 |
| | Rate the effect of water pollution when building a WtE Plant | 0 / 12 | 21 / 31 | 22.9 / 28 | 30 / 22 | 26 / 8 | 3.60 / 2.82 |
| | Rate the effect of land pollution when building a WtE Plant | 14 / 14 | 20 / 52 | 6 / 34 | 24 / 0 | 36 / 0 | 3.47 / 2.20 |
| | Rate the effect of noise pollution when building a WtE Plant | 6 / 9 | 3 / 34 | 17 / 31 | 0 / 22 | 74 / 5 | 4.34 / 2.78 |

**Table 6.** *Cont.*

| Social Impact Subcategory | Question/Indicator | Proportion (%) of Respondent Scores per Indicator for Likert Scale Scores (Highest Values Indicated in Bold) and Associated Band Score for Indicators from Numerical data | | | | | Subcategory Average Score |
|---|---|---|---|---|---|---|---|
| | | 1 | 2 | 3 | 4 | 5 | |
| | | Very Low | Low | Medium | High | Very High | Lagos Upper |
| | | | | | | | Abuja Lower |
| Location | Rate the effect of the proximity of a WtE plant on the public health of local residence. | 0 / 11 | 3 / 22 | 40 / 37 | 26 / 28 | 31 / 3 | 3.86 / 2.91 |
| | Rate the effect of the proximity of a WtE plant on the environment of the local residence. | 6 / 3 | 26 / 19 | 51 / 43 | 17 / 31 | 0 / 5 | 2.80 / 3.15 |
| | Rate the effect of the proximity of WtE plant on the sale or rent of properties of local residence. | 0 / 14 | 37 / 46 | 16 / 40 | 27 / 0 | 20 / 0 | 3.30 / 2.26 |
| | Rate the effect of the proximity of WtE plant on economic/commercial activities of local residence. | 0 / 3 | 27 / 19 | 41 / 31 | 20 / 31 | 11 / 17 | 3.16 / 3.40 |
| | Rate the effect of the proximity of WtE plant on job creation for local residence. | 0 / 5 | 14 / 6 | 31 / 28 | 44 / 28 | 10 / 34 | 3.50 / 3.80 |
| | Rate the level of comfort of having WtE plant within your vicinity | 23 / 12 | 27 / 12 | 13 / 34 | 3 / 22 | 34 / 20 | 2.99 / 3.25 |
| | How would rate managing waste as the most critical factor in the construction of a WtE Plant? | 20 / 5 | 23 / 6 | 4 / 22 | 0 / 40 | 53 / 28 | 3.70 / 3.80 |
| | How would rate electricity production as the most critical factor in the construction of WtE plant? | 23 / 6 | 0 / 9 | 0 / 19 | 27 / 40 | 50 / 26 | 3.81 / 3.71 |
| | How would rate environmental pollution is the most critical factor in the construction of WtE plant? | 17 / 11 | 26 / 9 | 0 / 31 | 11 / 28 | 46 / 22 | 3.43 / 3.40 |
| | How would you rate aesthetics as the most critical factor in the construction of a WtE plant? | 9 / 6 | 0 / 15 | 66 / 15 | 9 / 25 | 17 / 39 | 3.26 / 3.74 |
| | How would you rate local traffic burden as the most critical factor in the construction of a WtE plant? | 23 / 8 | 0 / 15 | 11 / 15 | 26 / 25 | 40 / 37 | 3.60 / 3.68 |
| Contribution to Economic Development | Rate the contribution WtE contribute to economic development | 0 / 6 | 9 / 15 | 6 / 15 | 39 / 25 | 47 / 39 | 4.24 / 3.74 |
| Public Acceptance | Rate the extent the public would accept the technology | 0 / 5 | 0 / 9 | 13 / 34 | 46 / 34 | 41 / 19 | 4.29 / 3.52 |

**Table 6.** *Cont.*

| Social Impact Subcategory | Question/Indicator | Proportion (%) of Respondent Scores per Indicator for Likert Scale Scores (Highest Values Indicated in Bold) and Associated Band Score for Indicators from Numerical data | | | | | |
|---|---|---|---|---|---|---|---|
| | | **1** | **2** | **3** | **4** | **5** | **Subcategory Average Score** |
| | | **Very Low** | **Low** | **Medium** | **High** | **Very High** | **Lagos Upper** |
| | | | | | | | **Abuja Lower** |
| Government Policy | How would you rate the strategies for the action of policies for waste management/WtE? | 0 9 | 3 12 | 23 49 | 16 25 | 59 5 | 4.30 3.03 |
| | How would rate adequacy of laws regulating waste management? | 0 11 | 11 28 | 29 52 | 24 6 | 36 3 | 3.84 2.63 |
| | How would you rate the strength of the waste management institutions? | 3 9 | 3 22 | 36 43 | 20 22 | 39 5 | 3.89 2.91 |
| Education and Training | How you rate the level of public education on waste management? | 0 9 | 4 40 | 57 31 | 19 12 | 20 8 | 3.54 2.69 |
| | Rate the level of training of waste workers | 0 14 | 9 40 | 30 28 | 23 19 | 39 0 | 3.91 2.51 |
| Improved Sanitation | Rate the level that WtE will increase the access to improved sanitation | 9 6 | 9 25 | 11 19 | 7 31 | 64 20 | 4.10 3.34 |
| | Rate the extent that WtE will improve the implementation of sanitation | 3 5 | 9 19 | 6 18 | 29 46 | 54 12 | 4.23 3.43 |
| | Rate the extent that WtE will encourage the participation in sanitation exercise | 7 0 | 0 0 | 33 31 | 11 52 | 49 17 | 3.94 3.86 |
| | Rate the extent the introduction of WtE will affect the percentage of payment for sanitation services | 3 8 | 3 22 | 11 28 | 27 40 | 56 3 | 4.30 3.09 |
| Improved Electricity Supply | Rate the extent the adoption of WtE will improve the electricity supply | 0 3 | 0 6 | 11 15 | 23 43 | 66 32 | 4.54 3.95 |
| Income | Rate the impact of WtE on disposable income of public servants | 0 20 | 9 46 | 34 25 | 19 9 | 39 0 | 3.87 2.23 |
| | Rate the impact of WtE on disposable income of consumers | 0 9 | 4 37 | 60 37 | 10 9 | 26 8 | 3.57 2.69 |
| | Rate the impact of WtE on paid living wage | 0 14 | 6 55 | 36 22 | 21 9 | 37 0 | 3.90 2.26 |
| | Rate the impact of WtE on regular payments | 0 6 | 13 43 | 14 28 | 36 15 | 37 8 | 3.97 2.75 |

**Table 7.** Scoring of Social Impact Indicators—numerical data assessment banded scores, [ ] = raw numerical data.

| Social Impact Subcategory | Indicator | Lagos | Abuja |
|---|---|---|---|
| Employment | Number of Jobs created | 3 [27,206 workers] | 2 [4536 workers] |
| Contribution of the Waste Management Sector to Economic Development | Percentage Contribution of Waste Management to the GDP of Nigeria (%) | 4 [3.75] | 4 [3.75] |
| | Amount of revenue generated annually from waste management (US$) | 2 [13,935,483 (in 2019)] | 1 [2,522,448 (in 2019)] |
| Improved Electricity Supply | Amount of Electricity Supplied Annually (kWh) | 3 [225,146,323 (in 2019)] | 4 [379,925,755 (in 2019)] |

**Table 8.** Social Impact subcategory scores for WtE in Lagos and Abuja (combined data from Tables 6 and 7) with Kruskal–Wallis test statistic. Figures are the mean scores, figures in parentheses = standard deviation (sd). Superscript numbering refers to the rank of the means.

| Social Impact Subcategory | Social Impact Scores | | | |
|---|---|---|---|---|
| | Lagos (n = 70) | Abuja (n = 65) | Combined | Kruskall-Wallis Test Statistic (df = 1) |
| Employment | $4.14^4$ (0.88) | $4.05^1$ (1.02) | $4.10^2$ (0.95) | 0.08 ns |
| Public Awareness | $3.73^8$ (1.58) | $2.74^9$ (0.89) | $3.25^8$ (1.38) | 16.17 *** |
| Health and Safety | $3.57^{10}$ (0.88) | $2.80^8$ (0.13) | $3.20^9$ (0.75) | 29.72 *** |
| Location | $3.40^{11}$ (0.55) | $3.37^6$ (0.24) | $3.39^7$ (0.43) | 2.86 ns |
| Contribution of Waste Management to Economic Development | $4.24^3$ (0.91) | $3.74^3$ (1.29) | $4.00^3$ (1.13) | 4.46 * |
| Public Acceptance | $4.29^2$ (0.68) | $3.52^4$ (1.05) | $3.92^4$ (0.99) | 19.95 *** |
| Government Policy | $4.01^6$ (0.43) | $2.86^7$ (0.41) | $3.45^6$ (0.72) | 87.16 *** |
| Education and Training | $3.73^8$ (0.30) | $2.60^{10}$ (0.24) | $3.19^{10}$ (0.63) | 105.96 *** |
| Improved Sanitation | $4.14^4$ (0.63) | $3.43^5$ (0.23) | $3.80^5$ (0.59) | 52.68 *** |
| Improved Electricity Supply | $4.54^1$ (0.69) | $3.95^2$ (1.01) | $4.26^1$ (0.91) | 14.76 *** |
| Income | $3.83^7$ (0.25) | $2.48^{11}$ (0.25) | $3.18^{11}$ (0.72) | 104.76 *** |
| Average Score | 3.97 (0.34) | 3.23 (0.56) | | |
| Kruskall–Wallis test statistic for subcategories between the two cities (df = 1) | 257.97 *** | | | |

ns = not significant at $p = 0.05$; * $p < 0.05$; *** $p < 0.001$. Colour coding based on aggregated overall Social Impact scores as:

| Very Negative Social Impact | Negative Social Impact | Neutral Social Impact | Positive Social Impact | Very Positive Social Impact |
|---|---|---|---|---|

*"Employment"* had the second-highest overall combined score, with the two cities having impact scores of 4.14 for Lagos (the 4th highest) and 4.05 (the highest by far) for Abuja, thereby indicating a strong recognition of the positive social impacts on employment that

adoption of WtE could provide. "*Contribution of Waste Management to Economic Development*" had the third-highest overall combined social impact score and was the most consistent subcategory for both Lagos and Abuja, indicating its strong relative importance for both cities. It is notable that 47% and 39% of respondents in Lagos and Abuja, respectively, which were the highest percentage of respondents for both cities, gave maximum scores of 5 (Very High) for the indicator "*Contribution of Waste Management to Economic Development*" subcategory (Table 6). "*Public Acceptance*" had the fourth-highest overall combined social impact score with Lagos (4.29) and Abuja (3.52) ranking it as the 2nd highest and 4th highest respectively. The indicator of this subcategory also revealed a consistency in terms of the views of respondents with 46% and 34% assigning a score of 4 (High).

The impact subcategories that had the lowest overall combined 'Social Impact' scores were "*Public Awareness*", "*Health and Safety*", "*Education and Training*", and "*Income*" with combined scores of 3.25, 3.20, 3.19, and 3.18, respectively. "*Public Awareness*" was placed 8th and 9th for Lagos and Abuja respectively although Lagos (3.73) had a higher impact score than Abuja (2.74). "*Health & Safety*" was 9th based on combined scores and 10th and 8th on individual scores for Lagos and Abuja, respectively. "*Income*" was at 11th place based on its overall combined score and had individual scores that placed it 7th and 11th for Lagos and Abuja respectively. The strikingly low ranking of "*Income*" as a Social Impact subcategory regarding WtE is notable, particularly from the individual indicator score for Abuja (Table 8). This comes from the largest number of respondents rating all the indicators of "*Income*" as Low by assigning a score of 2 (Table 6) and this was lower than the scores given by the Lagos respondents. "*Education and Training*" had a similar ranking between the two cities with Lagos and Abuja ranking it 8th and 10th, respectively.

The Social Impact subcategories "*Improved Sanitation*", "*Government Policy*" and "*Location*" were intermediate in scores ranking 5th, 6th, and 7th, respectively, in terms of overall score. Both "*Improved Sanitation*" and "*Government Policy*" were consistently ranked as intermediate by both cities achieving scores placing these subcategories as 4th (Lagos) and 5th (Abuja), and 6th (Lagos) and 7th (Abuja) out of the 11, respectively. For "*Location*", there was some heterogeneity between the two cities in their individual rankings with Lagos ranking it 11th and Abuja ranking it 6th.

Table 8 indicates positive scores in Lagos for all social impact subcategories except for the neutral score for "*Location*". There were no negative social impact scores for Lagos. This was not seen in the results for Abuja, where only four subcategories were perceived to have a positive social impact with six neutral and "*Income*" having a negative score. For the combined score, there were five subcategories exhibiting positive social impact with the remaining six subcategories showing a neutral social impact.

However, the "*Location*" subcategory was neutral for both cities. The fourth column in Table 8 gives the results from the Kruskal–Wallis non-parametric test applied to the data obtained for Lagos and Abuja only. This demonstrated statistically significant differences between the two cities for 9 of the 11 social impact subcategories.

For only two subcategories, "*Employment*" (positive social impact) and "*Location*" (neutral social impact), were the cities not significantly different. The Average Impact Score for both cities revealed a positive social impact for Lagos and a neutral social impact for Abuja with a clear significant difference ($p < 0.001$) between the two cities.

## 4. Discussion

This paper presents an assessment of the social impacts potentially arising from the adoption of WtE in Lagos and Abuja, Nigeria. The methodology was based on the UNEP sLCA guidelines and took a local participatory approach to identify key social impact issues and derive Social Impact subcategories relevant to the potential implementation of WtE in both locations. An indicator set was then constructed to calibrate the sLCA of the relative importance of the various Social Impact subcategories for WtE via a combination of interviews and a questionnaire-based survey with diverse local stakeholders in the two cities. The approach used is similar to that of [46,54], as applied to the promotion of the

Market Uptake for bio-based products from a consumer perspective and citrus farming, respectively, where the integration of participatory approaches such as the use of focus groups was identified as being important in making the sLCA more locally relevant.

Lagos and Abuja are the two major cities in Nigeria in terms of commercial activity and administration, respectively, and thus would be prime candidates for WtE if the Nigerian Government decided to adopt such technologies. But these cities have different socio-economic contexts that may well influence the arising social impacts and so the main aim of this research was to evaluate any differences in social impact perceptions between them. The focus groups resulted in the identification of 10 and 7 social issues for Lagos and Abuja, respectively—these shared some similarities but also differences. The main similarities were in the area of Contribution to Economic Development which were consistently ranked 3rd, in terms of the individual scores for both cities and the main differences were that Improved Electricity Supply was the most important issue in Lagos while Employment was the important issue in Abuja. Contrary to the study by Chong et al. [55], Land Use did not emerge as a key social issue for WtE implementation from our focus groups in both Lagos and Abuja. When these locally identified social issues were combined to generate a set of 11 Social Impact subcategories, four were in close alignment with those in the UNEP guidelines. However, the participatory process used here identified subcategories such as "*Improved Sanitation*" and "*Improved Electricity Supply*" that are not in the UNEP guidelines and this points to the need for, and supports, the flexibility within the guidelines regarding selection of subcategories for the assessment and its context (in our case WtE in two Nigerian cities). It should be noted that the present study was conducted with variable numbers of indicators to calibrate the different social impact subcategories (albeit with averaging and no weighting). The assumed use of equal weightings for the indicators within each sub-category is a point that needs to be explored in future work. Such methodological aspects in sLCA highlight its evolving nature and the fact that measuring social impacts in sLCA is not yet based on a fully elaborated and consensus theoretical grounding [56]. This perhaps could be considered as some of the limitations of this study where the use of scoring (and weighting) to assess social issues like this are based on a subjective assessment. This cannot however be avoided especially if adequate data is absent and quantification is required. As noted previously, this study is prospective rather than retrospective with uncertainties that attend preferences and perception over a future condition. In view of this, more research is still needed to develop characterization models that will allow for better aggregation and comparison of results which can be used to explain the suitable application of various systems/processes.

Regarding the use of participatory processes to derive subcategories and inform indicator choices, much can depend on who is interviewed and, in this case, on their knowledge of WtE. While the research aimed to provide an open opportunity for as many perspectives and issues to emerge, it is possible that some of the differences observed between the two cities may have been influenced by the composition of the focus groups and the interviewees and questionnaire respondents between the two cities, rather than being caused solely by the city contexts. Notwithstanding this, we consider that the degree of consistency between the cities in terms of sub-category rankings, etc., (even if the scores differed) does indicate that the approaches used were able to elucidate issues of local relevance and priority.

A comparison with previous sLCA studies, such as [16], shows that the present study provides insight on some new sLCA indicators and subcategories relevant to our context of Nigeria and WtE: these may well also be relevant in similar contexts. These subcategories were "*Improved Sanitation*" and "*Improved Electricity Supply*" along with their respective indicators. In terms of characterization of these and the other sLCA subcategories, the current study presented a useful way by which to convert qualitative data into quantitative data as well as the aggregation of these into comparable units. The methodology developed uses a Likert scale scoring system to assign scores to the indicators and subcategories. This made the proposed model somewhat simpler compared to other

LCIA assessment approaches, such as that proposed by Ciroth and Franze [53], which required expert judgement.

The highest overall ranking of the "*Improved Electricity Supply*" subcategory from the combined score for both cities (it however ranked 1st and 2nd individually in Lagos and Abuja, respectively) indicates that this may be expected to have the most positive social impact following the adoption of WtE. This finding agrees with those of Jekayinfa et al. [57], who noted that any additional supplement to the existing electricity supply (e.g., WtE) would be welcomed to improve electricity supply in Nigeria since demand is far more than the current 12 GW available via the national grid [58]. The "*Income*" subcategory, which may be expected to influence people's perception and attitude towards WtE [59], was ranked lowest in Abuja and in the combined scoring between both cities (it ranked 7th out of the 11 in Lagos). In addition to its ranking lowest of the 11 subcategories for Abuja, it was also the only social impact subcategory that indicated a negative social impact for the city. This negative impact could be a reflection towards those in the informal waste sectors (scavengers) whose sources of livelihoods depend on the sale of recyclable materials from the waste generated. If these waste fractions are then diverted to a WtE facility, the individuals would not have access to these materials and would lose their access to income [13]. However, further work is recommended to explore this matter in more detail.

In this research, the potential social benefits arising from the adoption of WtE were greater in Lagos than Abuja (with the possible exceptions of the "*Employment*" and "*Location*" subcategories). However, care needs to be taken here as the survey respondents were not asked to score the two cities, and respondents could only answer based on the knowledge they have of WtE and their own city. For instance, Table 3 depicts a higher percentage of the respondents in Lagos having university degrees as their highest educational qualification and earning a higher monthly income relative to those in Abuja. This could perhaps explain why significant differences in subcategories such as "*Public Awareness*" and "*Income*" could emerge from the respondents. But there are also possible explanations for some of the differences in impact subcategories between the cities. Differences in subcategories such as "*Public Acceptance*" may be attributed to the lack of public awareness and understanding of WtE in Abuja relative to Lagos [60], and a previously published study has noted that publicity of waste management activities, in general, is low in Abuja compared with Lagos [60].

Differences between the samples could also perhaps explain differences for the "*Health and Safety*" subcategory [61]. During the data collection process, the face-to-face interviews with key informants revealed that training opportunities in sustainable methods of MSW management are not readily available for operational staff in Abuja compared to Lagos and as a result waste management in Abuja is handled by less skilled staff and waste issues are decided mostly by political expediency rather than sound science [62]. Furthermore, findings from a study by Adenaike and Omotosho [63] reveal that the legislation and policies on waste management strategies in Lagos are strong compared to Abuja.

Finally, as Lagos has more industry and a larger population than Abuja, it is expected to have a higher demand for waste management and electricity than Abuja and, in turn, this could explain why subcategories such as "*Improved Electricity Supply*" and "*Improved Sanitation*" have higher scores in Lagos relative to Abuja. According to a study by Somorin et al. [64], Lagos has one of the highest electricity-generating potentials in the country while Abuja has one of the lowest.

Some of the findings from this study, especially with regards to "*Public Acceptance*", were consistent with those of other studies such as that by Achillas et al. [65], on the social acceptance for a WtE facility in Thessaloniki, Greece. Their results revealed a rather positive social impact from public attitudes on the integration of thermal treatment in the local waste management strategy and it was concluded that the public was not opposed to a WtE plant. A similar finding was obtained by Cucchiell et al. [66] when a social analysis through interviews was conducted to identify the most critical elements determining the

aversion toward WtE realization. This study discovered that, from a social perspective, the reduction in waste bills and a rigid and continuous control on emissions are able to support the realization of new plants but their implementation requires public approval. Another study on the social impacts of waste management strategies in Kabul, Afghanistan, in 2020 [16], showed a negative social impact on subcategories such as local employment and also indicated that subcategories such as child labor and working hours were the main social issues impacting waste management. This is somewhat different from the findings of the present study where WtE, as a prospective waste management strategy in Nigeria, had a positive social impact on "*Employment*", "*Improved Electricity Supply*" amongst others.

Our findings were consistent with those of Obidike et al. [67], who evaluated the socio-environmental and economic benefits of energy generation from MSW in Nigeria but using secondary data and tools such as content analysis, in-depth analysis, critical review, and narrative analysis. Obidike et al. [67] point out that WtE in Nigeria is affected by environmental policies, environmental education, and awareness, and the need to review Nigerian waste management policies, creation of environmental, economic awareness, and enlightenment was highlighted. These issues are very similar to those which emerged from the outcomes of our focus groups and which were brought into the sLCA as subcategories. The present study also adds further insight and support from our participatory work to the recent study conducted by Dunmade [68], which explored the social lifecycle assessment of bioenergy production from household and agricultural wastes in Lagos and Johannesburg. That study focused on evaluating the social lifecycle impacts of bioenergy facilities on workers, the consumers, the value chain, and the local community, and concluded that bioenergy could have positive impacts on employment, waste minimization, a cleaner environment, and improved communal health.

The sLCA conducted here was at the subcategory level with the transformation of both qualitative and quantitative indicators to enable the calibration and ranking of the subcategories. The benefit of analysis at this subcategory level is that it is possible to present a granular perspective on the best and worst performance in this sLCA by identifying the subcategories with the most or least impacts [69]. This 'bottom up' perspective from the subcategory development and calibration level was valuable in reflecting local perceptions and prioritization of social impacts between our two locations of study. An aggregation on the level of Impact Categories might well result in a loss of local relevance for the stakeholders as the 'higher level' of assessment may be too remote from the aspects that directly affect the stakeholders. This will be explored in our ongoing research.

The study here aimed to conduct a prospective sLCA with the integration of a participatory approach to identifying social issues of local relevance to stakeholders for the prospective implementation of WtE. We consider that this has been a useful approach, applicable to developing an overall understanding of the potential social impacts, and the acceptance of/enthusiasm for, WtE. The highlighting of different social impact perceptions between Lagos and Abuja for WtE provides valuable insight for decision-makers and stakeholders in delivering a sustainable path to tackle the twin problems of weak waste management and inadequate electricity supply.

The analyses performed here also link to wider perspectives in and beyond WtE by illustrating relevant context and approaches. For instance, D'Adamo et al.'s study [70] assessed the potential transition of the municipality of Rome to a more sustainable transport system and incorporated a "do nothing cost" as one of its indicators to measure the socio-economic impacts of the investment made. This enabled capturing of delay due to low public acceptance as well possible delay linked to uncertainties associated with the introduction of incentives. In studies relating to the food system and the circular economy, Giudice et al. [71] analyzed social impacts (as in the present research) to provide a useful social perspective framework for exploring the way alternative food systems can be connected with circular economy solutions such as WtE. The value of similar perspectives also is illustrated in the study by Alonso-Muñoz et al. [72], which examined how stakeholders in the supply chain are essential for reducing waste production and returning resources to the

production cycle in the implementation of circularity in organisations. This applicability of context can also be seen in other studies, such as another conducted by D'Adamo et al. [73] on the assessment of the recycling of end-of-life liquid crystal displays (LCDs) using a techno-economic analysis, and that conducted by Khan et al. [74], involving the optimal cost system of electricity generation for socio-economic sustainability in India.

The investigation of energy- and environment-related policies in this research has helped also to illustrate ways that issues relating to sustainable electricity generation can be addressed and how they are relevant to achieving several of the UN's Sustainable Development Goals (SDGs). For instance, at a very simple level, moving from fossil fuels to renewable energy sources is expected to help reduce carbon dioxide emissions and mitigate climate change. Discussion with some policymakers during the conduct of the research gave strong indications that the implementation of WtE would be a more environmentally friendly choice that should be integrated into policies such as the National Renewable Energy and Energy Efficiency Policy (NREEEP) as well as the National Policy on the Environment Regulations. This integration can enable the acceleration of an environmentally friendly transition in the energy sector. This integration can also serve as an opportunity for investments that will encourage local and foreign participation in the area of electricity generation [75]. However, this can be hindered by the non-implementation of these policies which is still a key issue in the Nigerian energy sector as indicated by Ogunmodimu and Okoroigwe [76]. Nonetheless, investigation of the other two dimensions of sustainability, namely the environmental and economic pillars, as part of a wider life cycle sustainability assessment (LCSA) is still needed as this will help guide the Nigerian government to devise appropriate policies for both sustainable waste management and electricity supply.

## 5. Conclusions

We conclude from this research that the adoption of WtE is expected to achieve higher overall levels of positive 'Social Impact' in Lagos compared with Abuja. The sLCA subcategory "*Improved Electricity Supply*" was ranked the highest both in Lagos and in the combined Lagos and Abuja scoring of prospective WtE adoption, while the sLCA subcategory "*Income*" was the least important in Abuja, (receiving a negative social impact score) and neutral overall in combined scoring for the two cities. Conducting the sLCA at the subcategory level has enabled such similarities and differences to be recognized in the analysis of social impact perceptions and priorities between the two cities in this study.

We also emphasize the value of the participatory processes in eliciting relevant issues and indicators when undertaking sLCA. The use of participatory approaches has been a key issue of this work in terms of its contribution to ongoing research on the sLCA method itself as it continues to develop further [23]. The acquisition of new knowledge in this area is supported to a great extent by the perspectives of various interest groups and stakeholders. This is important because stakeholders have significant roles, not only in identifying social issues that can affect the development of WtE in the contexts explored but also in providing their feedback on the use and development of sLCA itself, and by impacting policy development as the actors most affected by the system or process under consideration.

The sLCA methodology, and its ability to identify local specificities by involving experts and stakeholders, could clearly be adapted to other waste management processes by expanding the system boundaries and including additional relevant stakeholders. This implies that the approach and methodologies employed here can be replicated in other locations to identify and assess the social impacts of adopting WtE technology and other waste management strategies. Consequently, the results of this study provide suggestions and insights for policymakers in Nigeria and elsewhere for the development of WtE technologies as part of ongoing sustainable energy strategy development.

**Author Contributions:** Conceptualization, all authors; investigation, O.N.; data curation, all authors; writing—original draft preparation, all authors; writing—review and editing, all authors; visualization, all authors; supervision, S.M. and R.J.M. All authors have read and agreed to the published version of the manuscript.

**Funding:** This research was conducted as part of the first author's self-funded PhD research at the Centre for Environment & Sustainability, University of Surrey, UK.

**Institutional Review Board Statement:** The study was conducted according to the guidelines and procedures of the Ethics Committee of the University of Surrey, UK.

**Informed Consent Statement:** Informed consent was obtained from all subjects involved in the study.

**Data Availability Statement:** The data presented in this study are available on request from the corresponding author due to Confidentiality Issues.

**Acknowledgments:** The authors are most grateful to staff of Lagos Waste Management Authority and Abuja Environmental Protection Board for their assistance in data collection and to all members of the focus groups and stakeholder consultations and interviews.

**Conflicts of Interest:** The authors express no conflict of interest.

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
