# Peer review of "A Prospective Social Life Cycle Assessment (sLCA) of Electricity Generation from Municipal Solid Waste in Nigeria"

_sustainability, doi:10.3390/su131810177_

Round 1

Reviewer 1 Report

Dear authors,

my congratulations for this work. We have need to explore the relationship between sustainability and S-LCA. I ask only some details:

  1. Have you considered the last S-LCA? It is necessary to specify what is the exact point of reference.
  2. Section 2.1 is very long and it is not shows relevant informations
  3. Your analysis is well investigated but it is necessary also to show the potential "do nothing cost" https://doi.org/10.1016/j.renene.2020.10.072, the nexus circular economy-food https://doi.org/10.3390/su12197939 and the impact of supply chain on external factors https://doi.org/10.3390/su13116130
  4. Conclusions should be re-written. Logic flow and not a bullet point. What are the main issues of your work (e.g. different role of stakeholders involved), its replicability in other aspects.
  5. Discussion section: the impact of policies on this topic. 

Author Response

First and foremost, the authors would like to thank the three reviewers for their comments and suggestions. We are – of course – delighted to see that all three of them are very supportive of the work described in the paper.

Responses to the specific points raised by the reviewers are set out below; point raised by the reviewers are in black font while our responses are in red font. Most of the changes relate to the addition of new text so changes in the manuscript have been highlighted using red font (rather than track changes)

Reviewer 1

  1. My congratulations for this work.

Response: Thank you for this kind remark and support. It is much appreciated.

  1. I ask only some details: We have need to explore the relationship between sustainability and S-LCA

Response: Good point. We have inserted a brief explanation at line  85 in the revised manuscript. The text after that adds further explanation and we now explicitly refer the reader to section 2.2. where further detail of sLCA is presented in the Methods. We trust that this addresses the Reviewers point

  1. Have you considered the last S-LCA? It is necessary to specify what the exact point of reference is.

Response: We are not quite sure we have understood the point raised by the reviewer in this comment. We assume that the reviewer is inviting us to explain how our work builds on that of others in Nigeria and we have therefore sought to address it by referring to Dunmade’s (2019) work set-out in Line  626 of the revised manuscript

  1. Section 2.1 is very long and it is not shows relevant information

Response Thank you for this. It is a good point and we have shortened the text in a number of places in section 2.1. However, we do feel that it is important to give sufficient detail of the context of the two cities because of the comparative element of the research and their differences should therefore be made clear to the readership.

  1. Your analysis is well investigated but it is necessary also to show the potential "do nothing cost" https://doi.org/10.1016/j.renene.2020.10.072, the nexus circular economy-food https://doi.org/10.3390/su12197939 the impact of supply chain on external factors https://doi.org/10.3390/su13116130

Response:  Thank you for these valuable references. We have incorporated them into the revised text from line 656 onward in the revised manuscript.

  1. Conclusions should be re-written. Logic flow and not a bullet point. What are the main issues of your work (e.g. different role of stakeholders involved), its replicability in other aspects.

Response: Thank you for this suggestion which we have adopted by converting the bullets to a narrative text style and with additions from Line 705 in the revised manuscript.

  1. Discussion section: the impact of policies on this topic.

Response: Thank you for this suggestion. We agree with the reviewer that this is an important aspect which – while covered - was arguably under-represented in our original manuscript. We have added further explanation from Line 675 onwards to enhance this aspect in the Discussion.

Reviewer 2 Report

Overall it looks excellent. I have read a article carefully and I thought you might like a few suggestions.

  • Chapter should not end with a table.
  • As the given graphs must be visually clear, authors should redraw graphs 1; 2 as they are of poor quality.
  • The style of bibliography must be same.

Author Response

Reviewer 2

First and foremost, the authors would like to thank the three reviewers for their comments and suggestions. We are of course delighted to see that all three of them are very supportive of the work described in the paper.

Responses to the specific points raised by the reviewers are set out below; point raised by the reviewers are in black font while our responses are in red font. Most of the changes relate to the addition of new text so changes in the manuscript have been highlighted using red font (rather than track changes)

Overall it looks excellent. I have read a article carefully and I thought you might like a few suggestions.

Response: Thank you for the very positive comment on the paper. We are delighted to see this positive comment and the helpful suggestions

  1. Chapter should not end with a table.

Response: A very fair point and we have moved the last paragraph in Section 3.2 from before to after the table.

  1. As the given graphs must be visually clear, authors should redraw graphs 1; 2 as they are of poor quality.

Response: Fair point. The graphs have been redrawn and enlarged.

3. The style of bibliography must be same.

Response: Noted and we apologise. The issue has been addressed throughout the revised manuscript

Reviewer 3 Report

The article present an interesting and timely research. Overall the paper presentation is good. However, the following points observed

  • There are some minor typos and words repetition, this can be revised.
  • Some work related electricity generation can be added; for that see https://doi.org/10.1016/j.enpol.2019.01.013, https://doi.org/10.1016/j.tsep.2019.100390, https://doi.org/10.1016/j.enconman.2016.11.012 https://doi.org/10.3390/su13158256, https://doi.org/10.1007/s10668-020-01022-3 https://doi.org/10.1016/j.enconman.2016.11.012 https://doi.org/10.3390/su13158256, https://doi.org/10.1016/j.cogsc.2019.05.002
  • The conclusion section is scanty and must be improve, at this point the authors can mention the study limitation(s) and future scope.
  • What are the managerial insights and implications of the study. It can be explain for the policy makers to implement.
  • The sample questionnaire used in the study can be appended.

Best Wishes

Author Response

Reviewer 3

First and foremost, the authors would like to thank the three reviewers for their comments and suggestions. We are of course delighted to see that all three of them are very supportive of the work described in the paper.

Responses to the specific points raised by the reviewers are set out below; point raised by the reviewers are in black font while our responses are in red font. Most of the changes relate to the addition of new text so changes in the manuscript have been highlighted using red font (rather than track changes)

The article presents interesting and timely research. Overall the paper presentation is good.

Response: Thank you for this positive remark; it is much appreciated.

  1. However, the following points were observed. There are some minor typos and words repetition, this can be revised.

Response: Fair point and we apologize for this. We have given the paper a very thorough read through and have addressed the remaining typos throughout the revised manuscript

  1. Some work-related electricity generation can be added; for that see https://doi.org/10.1016/j.enpol.2019.01.013, https://doi.org/10.1016/j.tsep.2019.100390, https://doi.org/10.1016/j.enconman.2016.11.012, https://doi.org/10.3390/su13158256, https://doi.org/10.1007/s10668-020-01022-3, https://doi.org/10.1016/j.enconman.2016.11.012 https://doi.org/10.3390/su13158256, https://doi.org/10.1016/j.cogsc.2019.05.002

Response: Thank you for these suggestions. We have incorporated these references, for example at Lines 604, 685- 689 respectively.

3. The conclusion section is scanty and must be improved, at this point the authors can mention the study limitation(s) and future scope.

Response: Thank you for this point. It relates to a point raised by Reviewer #1 and we have addressed it by moving to a narrative style for the Conclusions (as per the recommendation of Reviewer 1) and the addition of text on limitation (s) and future scope is from line 513 -521 in the revised manuscript.

4. What are the managerial insights and implications of the study? It can be explained for the policymakers to implement.

Response: Thank you for this point. We have addressed it by adding text as of line 675.

5. The sample questionnaire used in the study can be appended.

Response: OK. The questionnaire used is has been appended as Supplementary Information

Round 2

Reviewer 1 Report

Congratulations

Reviewer 3 Report

The authors now have addressed my concerned